# Study on In-Doped CdMgTe Crystals Grown by a Modified Vertical Bridgman Method Using the ACRT Technique

**DOI:** 10.3390/ma12244236

**Published:** 2019-12-17

**Authors:** Pengfei Yu, Biru Jiang, Yongren Chen, Jiahong Zheng, Lijun Luan

**Affiliations:** 1School of Materials Science and Engineering, Chang’an University, Xi’an 710061, China; 2019131012@chd.edu.cn (B.J.); 2017131038@chd.edu.cn (Y.C.); jhzheng@chd.edu.cn (J.Z.); nmllj050@chd.edu.cn (L.L.); 2Engineering Research Central of Pavement Materials, Ministry of Education, Chang’an University, Xi’an 710064, China

**Keywords:** CdMgTe crystals, Te inclusions, annealing, crystal quality, radiation detector

## Abstract

Cadmium–magnesium–telluride (CdMgTe) crystal was regarded as a potential semiconductor material. In this paper, an indium-doped Cd_0.95_Mg_0.05_Te ingot with 30 mm diameter and 120 mm length grown by a modified Bridgman method with excess Te condition was developed for room temperature gamma-ray detection. Characterizations revealed that the as-grown Cd_0.95_Mg_0.05_Te crystals had a cubic zinc-blende structure and additionally Te-rich second phase existed in the crystals. From the tip to tail of the ingot, the density of Te inclusions was about 10^3^–10^5^ cm^−2^. The crystals had a suitable band-gap range from 1.52–1.54 eV. Both infrared (IR) transmittance and resistivity were relatively low. Photoluminescence measurement indicated that the ingot had more defects. Fortunately, after annealing, IR transmittance and the resistivity were significantly enhanced due to the elimination of Te inclusions. CdMgTe crystal after annealing showed a good crystal quality. The energy resolutions of the detector for ^241^Am and ^137^Cs gamma-ray were 12.7% and 8.6%, respectively. The mobility-lifetime product for electron was 1.66 × 10^−3^ cm^2^/V. Thus, this material could be used for room temperature radiation detectors.

## 1. Introduction

CdTe-based compound semiconductor materials are usually applied for optical isolators, Faraday rotators, photodetectors, and luminous diodes, especially promising room temperature radiation detectors because of their good optical and electrical properties [1,2,3]. Among II-VI semi-insulating semiconductors, CdZnTe is doubtlessly the most studied one. Recently, researchers pay more attention to developing new materials. Cadmium–magnesium–telluride (CdMgTe) as one of the potential radiation detection materials is highly regarded. In the last few decades, CdMgTe crystals were prepared for p-n junction electroluminescent diodes [4,5], mid-infrared solid-state lasers and passive optical Q-switches [6]. Until 2013, Hossain et al. [7] firstly reported the development of CdMgTe single crystal for X- and gamma-ray detectors. In comparison with CdZnTe, CdMgTe has some advantages, such as ideal energy band gap obtained by using a small amount of Mg in CdTe, appropriate segregation coefficient of Mg (almost 1.0) [8], and very close lattice constants of CdTe (6.48 Å) and MgTe (6.42 Å) [9]. Therefore, it is promising to grow large-size and homogeneous CdMgTe crystals with good crystallinity. The latest research for Cd_1–x_Mg_x_Te crystals grown by low-pressure Bridgman (LPB) technique was reported by Mycielski et al. [10]. But, the ingots had more defects. Up to now, few researches have reported on this material. It is necessary to develop large-size ingot for gamma-ray detectors.

According to the theory of deep energy level Te antisites [11], high-resistivity CdTe-based materials may be obtained by incorporating sufficient Te antisites and using indium as a shallow donor to compensate remainder Cd vacancies at the same time [12]. It is well known that the vertical Bridgman technique is a commonly easy and efficient method for crystal growth. So, in this work, a large-size Cd_0.95_Mg_0.05_Te ingot (30 mm diameter and 120 mm length) was successfully grown by a modified vertical Bridgman technique. Te-rich conditions, indium doping and the accelerated crucible rotation technique (ACRT) were used. Crystal structure, opto-electrical properties of CdMgTe were characterized. Post-growth annealing was also adopted for as-grown CdMgTe to improve crystal quality. Moreover, the performance of the detector was also measured.

## 2. Materials and Methods

High purity reagents, Cd (7 N), Mg (5 N), Te (7 N) and In (7 N), were applied to synthesize CdMgTe. A Cd_0.95_Mg_0.05_Te ingot was grown by the modified vertical Bridgman technique using the modifications of Te-rich condition, indium doping, and ACRT. For Te-rich condition, raw materials were weighed according to stoichiometric ratio, and 1.5 at.% excess Te was added to raw materials. The function of ACRT was to make the stabilization of growth velocity, the uniformity of component distribution, the control of the shape of liquid–solid interface, and the restraint of segregation [13]. Indium doping was at 10 ppm. The raw materials were sealed into a carbon film-coated quartz crucible. Then, the quartz crucible was vacuumed to 10^−5^ Pa. A rocking furnace was used to prepare CdMgTe polycrystals and the synthesis temperature was 1373 K. After that, CdMgTe ingot was grown in a two-zone furnace. The pulling rate and the temperature gradient were 1.0 mm/h and 10 K/cm, respectively. The quartz crucible was cooled to room temperature after crystal growth. Finally, we obtained a CdMgTe ingot with 30 mm diameter and 120 mm length, as shown in Figure 1. The as-grown (at the top part of Figure 1) ingot had a smooth surface with metallic sheen. After polishing (at the bottom of Figure 1), there were also no obvious voids in the surface of the ingot. The CdMgTe wafers were cut from parts of the tip, middle and tail along the axial direction. 5 × 5 × 2 mm^3^ sized small slices cut from the wafers were used for testing.

X-ray diffraction (AXS D8 ADVANCE X-ray diffractometer, Bruker, Billerica, MA, USA) was used to characterize crystal structure. The X-ray source was Cu Kα angular step was of 0.0001°. The energy band gap was testing by the near-infrared spectrum (SHIMADZU UV-3150 spectrometer, Shimadzu Corporation, Kyoto, Japan), and the wavelength range was from 600 to 1200 nm. IR transmission microscopy (Micronviewer 7290 A, American Electrophysics Company, Fairfield, NJ, USA) was adopted to observe Te-rich second phase. IR transmittance spectra (Nicolet Nexus 670 spectrometer, Thermo Fisher Scientific, Waltham, MA, USA) were measured, and the wavenumber range was from 4000 to 500 cm^−1^. Current–voltage (I-V) measurement (Agilent 4155 C instrument, Agilent Technologies, Santa Clara, CA, USA) was operated at room temperature. Photoluminescence spectra (Triax 550 tri-grating monochrometer, Horiba, Kyoto, Japan) were taken at the temperature as low as 10 K. ^241^Am and ^137^Cs gamma-ray energy spectra of the detector were measured on an ORTEC measurement system (Ortec, Oak Ridge, TN, USA) at room temperature.

## 3. Results

### 3.1. Crystal Structure

X-ray powder diffraction is used to determine the crystal structure of CdMgTe. Figure 2 shows X-ray diffraction (XRD) patterns of CdMgTe powders which come from different parts of the ingot. Obviously, the as-grown CdMgTe crystals have a cubic zinc-blende structure (JCPDS No. 15-0770) [14]. However, the crystals are not pure phase, because Te-rich second phase is observed. The crystal in the middle of the ingot has the lowest amount of Te-rich second phase. The production of abundant Te-rich phases should be the capture of a large number of Te-rich droplets due to the instability of liquid/solid interface during growth and desolventizing during the cooling process [15]. Moreover, the diffraction peak of (220) plane is the strongest instead of (111) plane, which exhibits the preferred orientation of crystal growth.

### 3.2. Te Inclusions in CdMgTe Crystals

CdTe-based crystals grown under stoichiometric [16] and excess Te conditions [17] commonly contain Te inclusions which existed as a main defect. The production of Te inclusion phase is mainly caused by the instability of liquid/solid interface and capturing of Te rich droplets during growth under supercooling [18]. IR 2-D images of CdMgTe crystals in different parts of the ingot are shown in Figure 3. Energy Dispersive X-ray (EDX) Spectroscopy analysis graph of a Te inclusion is also shown in Figure 3. It can be seen that Te inclusions have droplet shape in all crystals. This suggests that the inclusions are captured during the process of growth. The density of Te inclusions in the tip and tail parts is about 10^4^–10^5^ cm^−2^ order of magnitudes, which is more than that in the middle part (~10^3^ cm^−2^). The result is close to CdZnTe [19] and CdMnTe [20] crystals grown by the ACRT technique in our previous investigation. Differently, many large-sized inclusions from ten to over 100 microns were observed in the tip and tail parts of the CdMgTe ingot. We think that the main reason is the use of Te excess conditions (1.5 at.%). Due to the rather small amount of Mg, excess Te may become Te inclusions if the growth conditions are not well controlled. In fact, crystal growth is complicated, there are still many problems to be solved in order to obtain high-quality crystals.

As it is known, Te inclusions with high density and large size will seriously damage the crystal quality and deteriorate detector performance. Therefore, an effective annealing method [19] proposed in our previous research of CdMnTe crystal is adopted to eliminate Te inclusions in this paper. Figure 4 shows the IR photos of CdMgTe crystal before and after annealing. Obviously, Te inclusions with large sizes are significantly reduced after annealing under Cd atmosphere. Te inclusions with small size still exist in the crystal due to greater surface tension [21].

### 3.3. Energy Bang Gap 

Generally, the energy band gap of a semiconductor ranged from 1.5 to 2.2 eV is suitable for room temperature radiation detection. CdMgTe materials can reach an appropriate band gap by decreasing in Mg concentration because MgTe has a large band-gap of 3.5 eV [22]. An increment rate of the band-gap of about 17 meV per atomic percent Mg can be provided in CdMgTe crystal [23], compared with 13 meV per atomic percent Mn in CdMnTe and 6.7 meV per atomic percent Zn in CdZnTe [24]. Near infrared (NIR) transmittance spectrum is usually used to determine the energy band-gap of semiconductor. The test results of CdMgTe crystals cut from different positions of the ingot are shown in Figure 5.

It follows from Figure 5, that the wavelength is almost linear in the range of 750–850 nm. Therefore, the energy band gap is calculated by the formula [25]
(1)$$\alpha =\frac{B{\left(h\nu -E_{g}\right)}^{\gamma }}{h\nu },$$ where *α* is the absorption coefficient, *hν* is photon energy, *B* is the proportional constant, *E_g_* expresses the energy band gap and *γ* is the index of the mechanism of the electron transmitting from valence band to conducting band. For direct band-gap semiconductors, *γ* is $\frac{1}{2}$. Then, the Equation (1) is expressed as
(*αhν*)^2^ ∝ (*hν* − *E*_g_),(2) where (*αhν*)^2^ has a linear relationship with *hν*. The values of *E*_g_ can be determined by extrapolation method to about 1.54 eV, 1.53 eV and 1.52 eV for the tip, middle and tail of the ingot, respectively (see the inset in Figure 5). It shows that Mg concentration is relatively homogeneous. An appropriate band-gap can be achieved when the concentration of Mg is 5%. This result is close to that of Cd_0.9_Zn_0.1_Te, and is also approximately close to the value reported by Hossain et al. [7].

### 3.4. Electrical Property

Before testing electrical property, ohmic contacts are formed by depositing circular gold electrodes of 2 mm in diameter on both sides of the CdMgTe crystals. Figure 6 shows the I-V curves of CdMgTe crystals cut from different positions of the ingot. The resistivities of the tip, middle and tail of the crystals are about 3.15 × 10^6^ Ω·cm, 2.34 × 10^7^ Ω·cm and 6.71 × 10^5^ Ω·cm, respectively (Figure 6a). Obviously, the resistivity of the whole ingot is low. We think that the II–VI group compound semiconductor has a self-compensation effect related to complexes of $In_{\mathrm{Cd}}^{+}$ and $V_{\mathrm{Cd}}^{2-}$ [26]. In this case, the use of indium as a donor may inefficiently compensate the remainder of Cd vacancies after the compensation of TeCd. Moreover, the measured concentration of Te inclusions shows that TeCd as a donor in CdMgTe ingot is excess. This also results in the low resistivity for CdMgTe crystals. We all know that high resistivity can ensure low leakage current, thus reducing the electrical noise of the detector. The resistivity is a significant factor for room temperature radiation detection. So, post-growth annealing [19] for CdMgTe is adopted to improve resistivity to satisfy the fabrication of the detector. Hall measurement indicates that the crystal after Cd atmosphere annealing has n-type conduct with low resistivity. Then, the resistivity of CdMgTe crystal is enhanced by the subsequent Te atmosphere annealing. The crystal becomes weak n-type conduct. The resistivity is 5.77 × 10^10^ Ω·cm after annealing (Figure 6b), which is approximately four orders of magnitudes improvement over as-grown crystal. This is a qualitative change for as-grown crystal. The significant improvement for the resistivity is ascribed to the deep energy level Te antisites, which is able to pin the Fermi level to the middle of the band [27].

### 3.5. Optical Property

IR transmittance is usually related to crystal quality [28]. IR transmittance spectra of CdMgTe crystals in different parts of the ingot are shown in Figure 7. When the wave number is between 500 cm^−1^ to 4000 cm^−1^, the average transmittances of the tip, middle and tail of the ingot are about 33%, 45% and 24%, respectively (Figure 7a). Low transmittance suggests that the ingot does not have good crystal quality. This is because the crystals have low resistivity and high-density inclusions. But, the crystal quality can be improved after annealing. The average transmittance of the crystal in the middle part of the ingot is enhanced to 57% after annealing, with a 12% increase (Figure 7b).

Photoluminescence (PL) spectroscopy is commonly used to characterize point defects in CdTe-based crystals. Figure 8 shows the PL spectra of CdMgTe crystals at the temperature of 10 K. It is similar to CdZnTe crystal [29] that the PL spectra of CdMgTe crystals can be divided into three main regions: (1) near-band-edge region ranged from 1.67 to 1.71 eV, (2) donor-acceptor pair (D, A) region ranged from 1.58 to 1.67 eV, and (3) A-center region ranged from 1.40 to 1.58 eV. 

For as-grown crystals (Figure 8a), in the region (1), the intensity of (D^0^, X) peak representing neutral donor bound exciton is weak, which suggests that CdMgTe crystals do not possess good crystal quality. This peak almost disappears in the tail crystal. In the region (2), donor-acceptor pair (DAP) peak related to impurities appears. In the region (3), a peak is named as D_complex_. The intensity of D_complex_ is high for all crystals. In CdZnTe: In [30] and CdMnTe: In [17], D_complex_ peak can be attributed to indium and Cd vacancy associated complexes and dislocation. Thus, for CdMgTe crystal, D_complex_ peak is also due to indium impurity and dislocation produced by Te inclusions. Because Te inclusions are apparently reduced after annealing, the intensity of (D^0^, X) peak has a prominent increase for the annealed CdMgTe crystal in Figure 8b.

### 3.6. Detector Performance

The detector can be fabricated using the annealed crystal due to the great enhancement of the resistivity. Before testing energy spectra, Au coating was used as the electrode material to prepare the planar detector. The ^241^Am (59.5 keV) and ^137^Cs (662 keV) gamma-ray spectra of the detector are shown in Figure 9. The energy resolutions of the detector for ^241^Am and ^137^Cs gamma-ray are 12.7% and 8.6%, respectively, when 420 V bias voltage and 1 μs shaping time are applied.

Then, the collection efficiency of the detector is determined by applying different bias voltages. Therefore, (μτ)_e_ expressed as the mobility lifetime product for electron can be calculated according to the Hecht equation [31]. This value is 1.66 × 10^−3^ cm^2^/V in this work (see Figure 10). The result is close to CdZnTe [32] and CdMnTe [20]. It suggests that CdMgTe crystal is a potential material to be used for a room temperature radiation detector.

## 4. Conclusions

CdMgTe crystal was a promising room temperature radiation detection material due to some potential advantages. An indium-doped Cd_0.95_Mg_0.05_Te ingot with 30 mm diameter and over 100 mm length was successfully grown by a modified Bridgman method with excess Te. The crystals were measured by XRD analysis, IR image, NIR spectrum, I-V curve, IR transmittance, and PL spectrum. The results indicated that the as-grown Cd_0.95_Mg_0.05_Te crystals had a cubic zinc-blende structure. And Te-rich second phase existed in the crystals. From the tip to tail of the ingot, the density for Te inclusions was about 10^3^–10^5^ cm^−2^ orders of magnitudes and the energy band gap of CdMgTe crystals was 1.52 eV to 1.54 eV. IR transmittance of the whole ingot was lower than 45%. The resistivity was 2.34 × 10^7^ Ω·cm to 6.71 × 10^5^ Ω·cm. In PL measurement, the defect luminescence peak with high intensity suggested that the ingot did not have good crystal quality. After annealing, Te inclusions were significantly eliminated. IR transmittance and the resistivity were increased by 12% and approximately 4 orders of magnitudes, respectively. (D^0^, X) peak with high intensity in PL spectrum showed a good crystal quality for the annealed CdMgTe crystal. The detector fabricated by the annealed crystal could be used for room temperature radiation detection.

## Figures and Tables

**Figure 1 materials-12-04236-f001:**
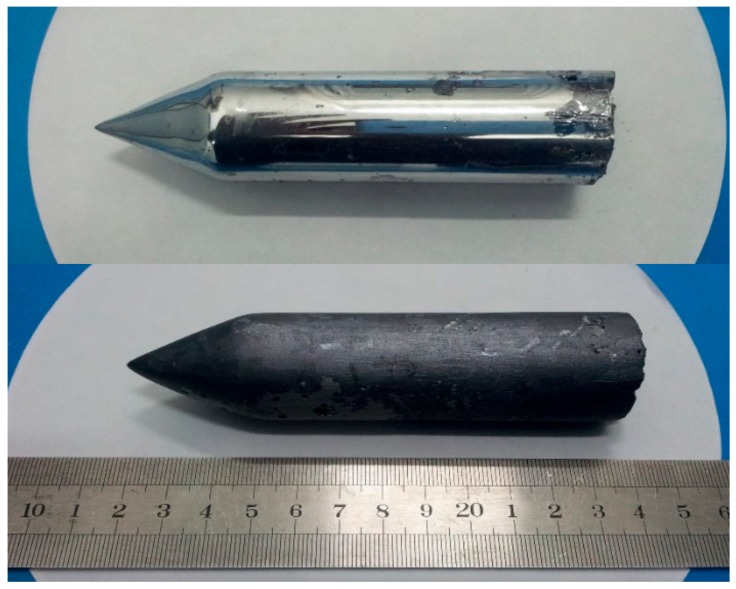
Picture of the cadmium–magnesium–telluride (CdMgTe) ingot grown by the modified vertical Bridgman method.

**Figure 2 materials-12-04236-f002:**
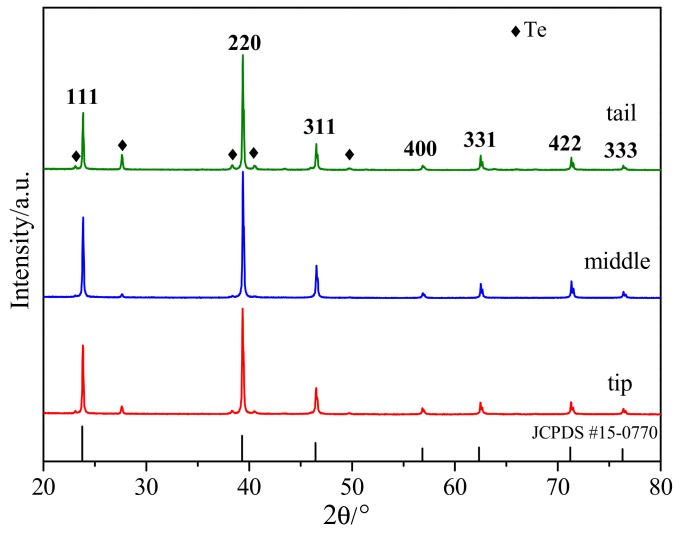
X-ray diffraction (XRD) patterns of CdMgTe powders in different parts of the ingot.

**Figure 3 materials-12-04236-f003:**
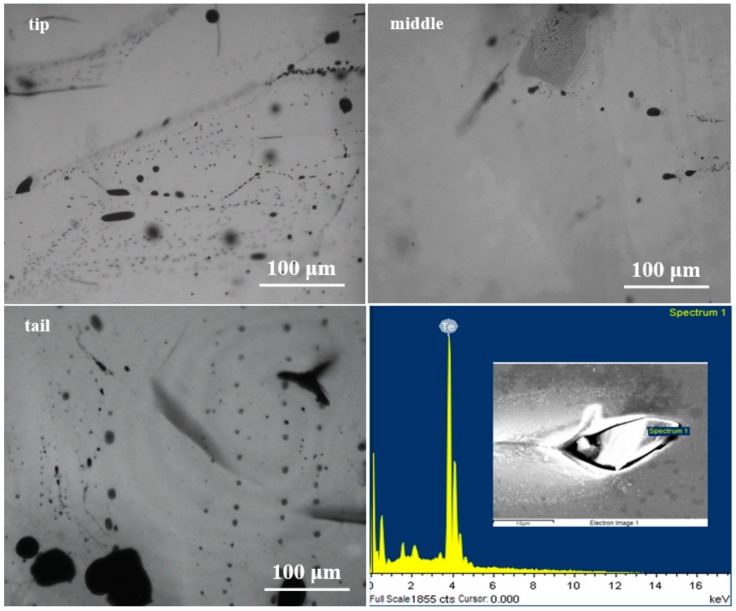
IR images of CdMgTe crystals in different parts of the ingot and EDX analysis image of a Te inclusion. Tip, middle and tail represent the positions of the crystals in the ingot. Spectrum 1 represent the point of EDX measurement.

**Figure 4 materials-12-04236-f004:**
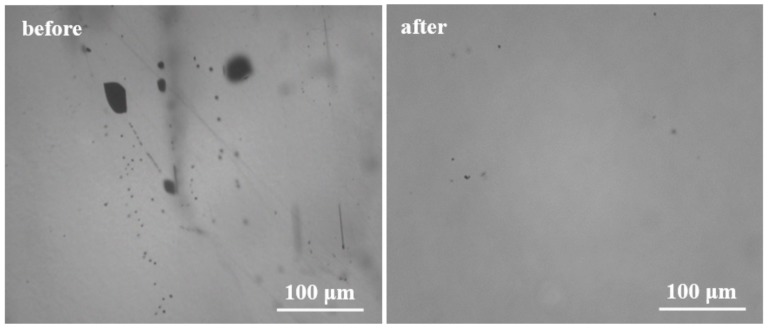
IR images of CdMgTe crystal before and after annealing.

**Figure 5 materials-12-04236-f005:**
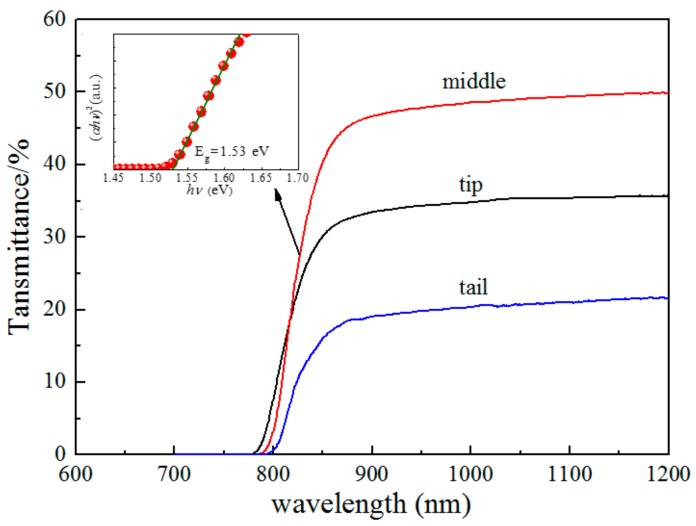
Near infrared (NIR) transmittance spectra of CdMgTe crystals in different parts of the ingot. The inset is the fitting results of (*αhν*)^2^ and *hν*.

**Figure 6 materials-12-04236-f006:**
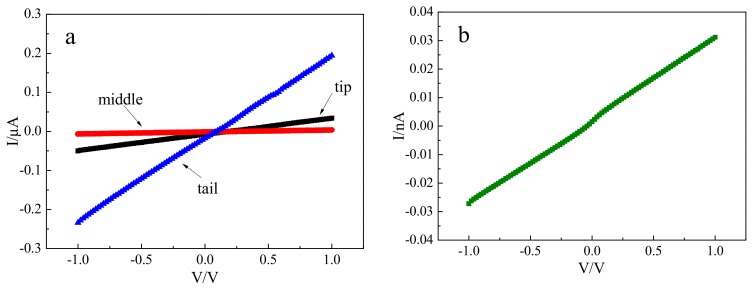
I-V curves of CdMgTe crystals: (**a**) as-grown; (**b**) after annealing.

**Figure 7 materials-12-04236-f007:**
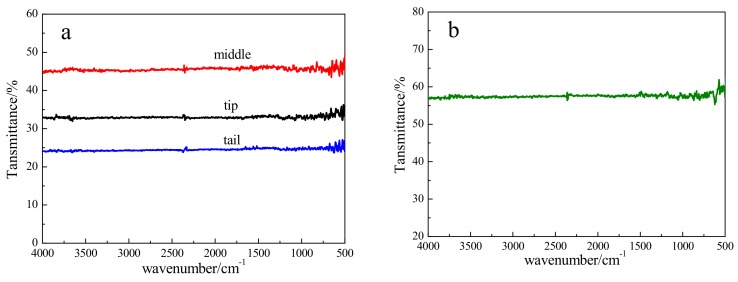
IR transmittance spectra of CdMgTe crystals: (**a**) as-grown; (**b**) after annealing.

**Figure 8 materials-12-04236-f008:**
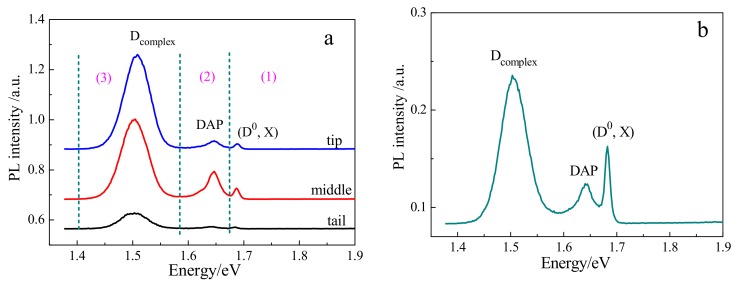
Photoluminescence (PL) spectra of CdMgTe crystals in different parts of the ingot: (**a**) as-grown; and (**b**) after annealing.

**Figure 9 materials-12-04236-f009:**
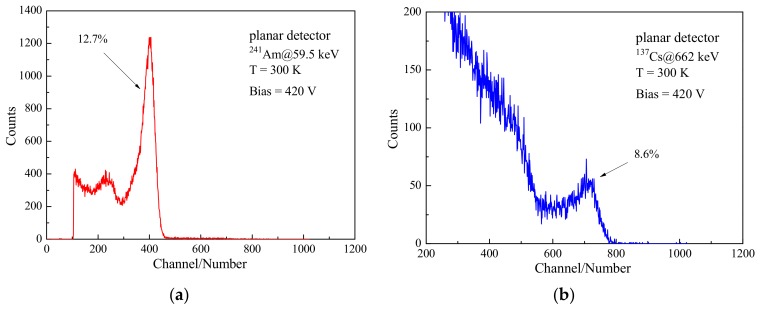
Gamma-ray spectra of CdMgTe detector: (**a**) ^241^Am; and (**b**) ^137^Cs.

**Figure 10 materials-12-04236-f010:**
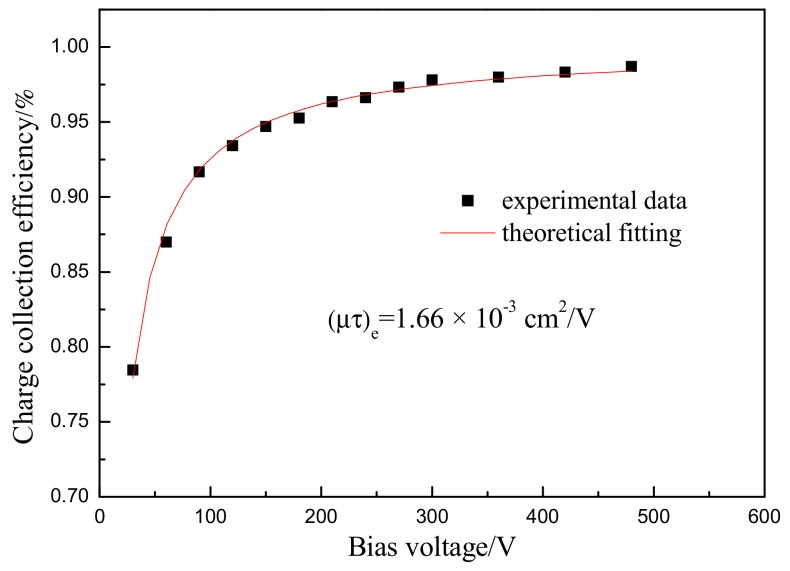
Collection efficiency of CdMgTe detector.

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
