# Peer review of "Study on In-Doped CdMgTe Crystals Grown by a Modified Vertical Bridgman Method Using the ACRT Technique"

_materials, 2019, doi:10.3390/ma12244236_

Round 1

Reviewer 1 Report

On the whole the work is interesting, the results seem reliable, disscussion is reasonable and supported by the experiment. But English is bad, and it spoils all meaning of the work. IMPROVE it or otherwise I cannot recommend your paper for publishing.

As to the results.

How much In did you use for doping? It seems to be a small amount, but it is not clear from the text. You say that you see Te-rich phase in diffraction patterns, can you identify this phase? Write what irradiation was used in XRD. If you tell about your samples having a crystallographic texture based on intensity of XRD peaks, you should compare relative intensities of experimental diffraction pattern and those in the JCPDS database. At which temperature precisely were taken PL spectra? The text with my corrections is enclosed. Follow them and be aware that you should reread the corrected text properly.

Author Response

Point 1: On the whole the work is interesting, the results seem reliable, disscussion is reasonable and supported by the experiment. But English is bad, and it spoils all meaning of the work. Improve it or otherwise I cannot recommend your paper for publishing. Response 1: According to the reviewer’s comments, we have revised our manuscript carefully to improve English. Point 2: How much In did you use for doping? It seems to be a small amount, but it is not clear from the text. You say that you see Te-rich phase in diffraction patterns, can you identify this phase? Write what irradiation was used in XRD. If you tell about your samples having a crystallographic texture based on intensity of XRD peaks, you should compare relative intensities of experimental diffraction pattern and those in the JCPDS database. At which temperature precisely were taken PL spectra? The text with my corrections is enclosed. Follow them and be aware that you should reread the corrected textproperly. Response 2: The concentration of Indium was 10 ppm, you can see in the part of “Materials and Methods”. For Te-rich phase, XRD measurements indicate that it exists in the CdMgTe crystals (see Figure 2). Tip and tail of the ingot have more Te-rich phase. Characteristic XRD peak of Te-rich phase is observed obviously and marked. Moreover, we have identified a Te inclusion by SEM (see Figure 3). We have added the irradiation used in XRD, you can see in the part of “Materials and Methods”. We have made the XRD patterns of CdMgTe powders again and added the JCPDS database. The precise temperature of PL spectra is 10 K, you can see in the manuscript. According to the reviewer’s corrections, we have revised our manuscript carefully. Thanks very much for reviewer’s suggestions. The revision is red and can be seen in the article.

Reviewer 2 Report

The paper deals with the growth and characterization of CdMgTe and the optimization of this material for radiation detection. The investigated material represents promising option, how researchers could extend
their effort and the presented research in interesting. Authors proved that they can prepare detector, which may be used at some less demanding applications. My big concern is related with the used growth method - ACRT with using Te excess. The paper could be published in Materials after authors satisfactorily explain my comment and questions below.

1. The growth of CdTe-related single crystals from the Te-rich melt using ACRT was successfully used in some laboratories. They proved that ACRT may effectively suppress the formation of Te inclusions. In spite of using ACRT, huge inclusions are detected in as-grown crystals in this manuscript. There appears a question, what is the reason of this result. Was it the rather small amount of Mg? More probably, I guess, the ACRT was not mastered in the authors' laboratory and the excessive Te played the role as it would be even in the pure CdTe. If such a case was true, why authors used their technique requesting subsequent annealing instead of obvious Bridgman or other techniques producing high resistivity materials without the follow-up processing? Could authors compare inclusions in their CdMgTe crystals with CdTe or CdZnTe grown by the same technique ACRT? It is known that the follow-up annealing cannot remove all defects induced by inclusions and the quality of detectors remains reduced. I thus conclude that the reported research could be hardly used as a recommendation to researchers to follow presented growth method and the importance of the manuscript for utilization by researchers is low from this point of view. If authors are of another belief, I will eagerly read their arguments.

2. The Cd annealing leads generally to the dissolution of inclusions but simultaneously low resistivity n-type is obtained. Authors should confirm, if it was the same also in their samples.

3. The title should better inform on the growth method ACRT.

4. English should be improved. Multiple drawbacks were detected.

Author Response

Point 1: The growth of CdTe-related single crystals from the Te-rich melt using ACRT was successfully used in some laboratories. They proved that ACRT may effectively suppress the formation of Te inclusions. In spite of using ACRT, huge inclusions are detected in as-grown crystals in this manuscript. There appears a question, what is the reason of this result. Was it the rather small amount of Mg? More probably, I guess, the ACRT was not mastered in the authors' laboratory and the excessive Te played the role as it would be even in the pure CdTe. If such a case was true, why authors used their technique requesting subsequent annealing instead of obvious Bridgman or other techniques producing high resistivity materials without the follow-up processing? Could authors compare inclusions in their CdMgTe crystals with CdTe or CdZnTe grown by the same technique ACRT? It is known that the follow-up annealing cannot remove all defects induced by inclusions and the quality of detectors remains reduced. I thus conclude that the reported research could be hardly used as a recommendation to researchers to follow presented growth method and the importance of the manuscript for utilization by researchers is low from this point of view. If authors are of another belief, I will eagerly read their arguments.

Response 1: We agree with the reviewer. ACRT may effectively suppress the formation of Te inclusions in some researches. Generally, ACRT is used to make the stabilization of growth velocity, the uniformity of component distribution, the control of the shape of liquid-solid interface, and the restraint of segregation. In fact, crystal growth is complicated, and many defects will appear if the growth conditions are not well controlled. In our research, one of the functions of ACRT is to suppress Te inclusions. But, many and huge inclusions exist in the ingot. Another function of ACRT is to uniform Mg distribution. NIR measurements indicate that the distribution of Mg element is homogeneous from the tip to tail. So, we suspect that the main reason is the use of Te excess conditions. Excess Te is usually used in the growth of Cd0.9Zn0.1Te (P. Yu, et al. Nuclear Instruments and Methods in Physics Research A, 2014, 737: 29–32) and Cd0.9Mn0.1Te (Y. Du, et al. Journal of Crystal Growth, 2011, 318: 1062-1066). The density of Te inclusions in our previous research for both CdZnTe and CdMnTe crystals grown using ACRT is about 104 cm-3. The result is close to the density of Te inclusions in CdMgTe. The difference is that there are huge Te inclusions in this study. Due to rather small amount of Mg, excess Te (1.5 at.%) may become Te inclusions in this paper although using ACRT. This is the first time we have grown CdMgTe crystals. There are still many problems to be solved in order to grow high-quality crystals. Aware of the problem of many and huge inclusions, we will improve the growth conditions in subsequent experiments. Because there are many defects, we anneal the as-grown crystals. The best result, of course, is to grow high-quality single crystals. That is what we are trying to do. Many thanks to the reviewers for pointing out the shortcomings of our research.

Point 2: The Cd annealing leads generally to the dissolution of inclusions but simultaneously low resistivity n-type is obtained. Authors should confirm, if it was the same also in their samples.

Response 2:  Thanks for the reviewer’s comments. We have confirmed the conductive type of the crystal after annealing. Hall measurement indicates that the crystal after Cd atmosphere annealing has n-type conduction with low resistivity. Then, the resistivity of CdMgTe crystal is enhanced by the subsequent Te atmosphere annealing. The crystal becomes weak n-type conduction. The revision is red and can be seen in the article.

Point 3: The title should better inform on the growth method ACRT.

Response 3: According to the reviewer’s comments, we have added ACRT to the title. The revision is red and can be seen in the article.

Point 4: English should be improved. Multiple drawbacks were detected.

Response 4: According to the reviewer’s comments, we have carefully revised our manuscript to improve English.

Round 2

Reviewer 1 Report

The article is now readable, it is of interest for specialists, but still some words are inappropriate. They are out of place and perplex a reader. I made some minor corrections. Please, make them before publishing.

Author Response

Point 1: They are out of place and perplex a reader. I made some minor corrections. Please, make them before publishing.

Response 1: According to the reviewer’s comments, we have revised the manuscript. Thanks a lot for reviewer’ suggestion!

Reviewer 2 Report

Authors amended the manuscript according my recommendation. I approve its publishing in Materials.

Author Response

Thanks a lot for reviewer’ comments!